# A Non-Disposable Electrochemical Sensor Based on Laser-Synthesized Pd/LIG Nanocomposite-Modified Screen-Printed Electrodes for the Detection of H_2_O_2_

**DOI:** 10.3390/s24072043

**Published:** 2024-03-22

**Authors:** Ruijie Song, Jianwei Zhang, Ge Yang, Yu Wu, Jun Yu, Huichao Zhu

**Affiliations:** 1Department of the School of Medicine, Dalian University of Technology, Dalian 116024, China; songruijie@mail.dlut.edu.cn (R.S.); g32109306@mail.dlut.edu.cn (G.Y.); wyrose@mail.dlut.edu.cn (Y.W.); junyu@dlut.edu.cn (J.Y.); 2Department of the School of Control Science and Engineering, Dalian University of Technology, Dalian 116024, China; jwzhang@dlut.edu.cn

**Keywords:** nanoparticles, laser-induced graphene, hydrogen peroxide, electrochemical detection

## Abstract

There have been many studies on the significant correlation between the hydrogen peroxide content of different tissues or cells in the human body and the risk of disease, so the preparation of biosensors for detecting hydrogen peroxide concentration has been a hot topic for researchers. In this paper, palladium nanoparticles (PdNPs) and laser-induced graphene (LIG) were prepared by liquid-phase pulsed laser ablation and laser-induced technology, respectively. The complexes were prepared by stirring and used for the modification of screen-printed electrodes to develop a non-enzymatic hydrogen peroxide biosensor that is low cost and mass preparable. The PdNPs prepared with anhydrous ethanol as a solvent have a uniform particle size distribution. The LIG prepared by laser direct writing has good electrical conductivity, and its loose porous structure provides more adsorption sites. The electrochemical properties of the modified electrode were characterized by cyclic voltammetry, chronoamperometry, and electrochemical impedance spectroscopy. Compared with bare screen-printed electrodes, the modified electrodes are more sensitive for the detection of hydrogen peroxide. The sensor has a linear response range of 5 µM–0.9 mM and 0.9 mM–5 mM. The limit of detection is 0.37 µM. The above conclusions indicate that the hydrogen peroxide electrochemical biosensor prepared in this paper has great advantages and potential in electrochemical catalysis.

## 1. Introduction

Hydrogen peroxide (H_2_O_2_) is a kind of inorganic compound with strong oxidizing and reducing properties. Due to its unique physical and chemical properties, hydrogen peroxide is widely used in industrial production. For example, it can be used as a bleach in the papermaking and textile processes [1], and can also be used as a strong oxidant in the preparation of cleaning products and in disinfection [2,3]. In addition, hydrogen peroxide can react with other compounds to solve environmental problems such as organic wastewater. At the same time, hydrogen peroxide is also an essential compound for maintaining physiological balance in organisms, and plays an important role in the cascade of biological signals, such as cell membrane signal transmission, cell differentiation, cell apoptosis, and gene expression [4]. Hydrogen peroxide is a by-product of several enzymatic reactions produced by every cell in the body. Its normal concentration is in the range of 0.001–0.1 µM, and elevated concentrations may lead to increased susceptibility to disease in the body. Hydrogen peroxide exhibits different biological properties at different sites of accumulation in the body, resulting in different clinical manifestations. At the cellular level, accumulation of hydrogen peroxide triggers apoptosis, leading to autoimmune diseases. At the tissue level, accumulation of hydrogen peroxide leads to inflammation and ulceration. At the systemic level, toxic concentrations of hydrogen peroxide in the blood can cause organ failure and microvascular dysfunction in the body [5,6]. Therefore, an accurate, sensitive, and rapid method for the determination of H_2_O_2_ concentration is of great significance, especially in medical and industrial fields. Various methods have been used for H_2_O_2_ detection, such as titration [7], chromatography [8], fluorescence [9], spectrophotometry [10], an electrochemical method [11], etc. However, some of these detection methods are limited due to their complex operation and high cost. The detection of H_2_O_2_ by the electrochemical method is low cost, easy to operate, and has better sensing parameters. The miniaturization and modifiability of the electrodes make it possible to detect H_2_O_2_ in a variety of environments [12,13,14].

Currently, non-enzymatic electrochemical sensors are the main way to detect H_2_O_2_ because of their long lifetime and good stability. Graphene is receiving more and more attention due to its excellent conductivity and large specific surface area, which can enable electrochemical sensors with excellent detection capabilities. Graphene is a kind of hexagonal honeycomb lattice composed of carbon atoms with sp^2^ hybridized orbitals, and its thickness is only the size of an atom, which makes it a two-dimensional material [15]. Its electron mobility exceeds 15,000 cm^2^/V·s at room temperature, and its resistivity is about 1 × 10^−6^ Ω·cm, which makes it the lowest resistivity material in the world [16]. Graphene has been widely used in biofuel cells, supercapacitors, and flexible sensors, and for biodetection [17,18,19,20]. Since the mid-20th century, a number of methods for preparing graphene have emerged, which are mainly categorized into top-down exfoliation and bottom-up growth. For example, micromechanical exfoliation [21], chemical vapor deposition [22], epitaxial growth [23], the redox method [24], and organic synthesis [25] have been successfully applied. Although these traditional preparation methods are very mature, there are still some problems such as complex operation and chemical hazards, so the emergence of laser-induced graphene technology has set off a wave of research. In 2014, Tour’s group used a CO_2_ infrared laser as a light source to prepare porous laser-induced graphene (LIG) using an organic material film, polyimide film, as a precursor [26]. This is the first time that researchers have obtained graphene by means of laser irradiation. Using this graphene production method, graphite structures with customized properties can be synthesized from different precursor materials, such as synthetic polymer materials like polyimide, natural materials like wood and paper, etc., which greatly reduces the difficulty of production compared to traditional processes. However, the catalytic ability of LIG by itself is not powerful enough, and it is often necessary to prepare catalytic materials by complexing them with noble metal nanoparticles, which is an effective way to significantly improve the performance of the sensors and has already been applied [27,28,29,30]. Among the precious metals, palladium nanoparticles (PdNPs) have become one of the most commonly used catalysts for electrochemical sensors due to their relatively low price and excellent catalytic ability.

In this work, we synthesized graphene composites with precious metal nanoparticles by laser direct writing technology and liquid-phase pulsed laser ablation techniques. As far as we know, no one has prepared nanocomposite materials by laser for the detection of hydrogen peroxide. At present, most electrochemical biosensors based on LIG are patterned directly during irradiation, then the formed three electrodes are modified and directly used for substance detection. However, a major disadvantage of this preparation method is that it requires encapsulation treatment to establish hydrophilic and hydrophobic regions, and such sensors cannot be reused. The sensor designed in this work can be utilized several times, which helps to reduce the cost of detection. The structure and morphology of the prepared materials were studied by scanning electron microscopy (SEM) and Raman spectroscopy. The sensitivity, detection limit, and linear range of the modified electrode were studied by cyclic voltammetry (CV), chronoamperometry (IT), and electrochemical impedance spectroscopy (EIS). In addition, the stability, reproducibility, and anti-interference of the sensor were evaluated.

## 2. Materials and Methods

### 2.1. Reagents, Chemicals, and Instruments

Anhydrous ethanol (C_2_H_5_OH), hydrogen peroxide (H_2_O_2_,35%), acetone (C_3_H_6_O), phosphate buffer solution (PBS, pH = 7.0), sodium tetraborate (Na_2_B_4_O_7_), potassium ferricyanide (K_3_Fe(CN)_6_), ascorbic acid (AA), dopamine (DA), uric acid (UA), glucose (Glu), sodium chloride (NaCl), and potassium chloride (KCl) were purchased from China National Medicines Corporation Ltd(Beijing, China) at the highest available purity. The 783 solvent was purchased from China Dongguan Lijing Printing Materials(Dongguan, China) at chemical purity. CH-8 conductive carbon paste (MOD2) purchased from JUJO Chemical Corporation Ltd (Tokyo, Japan) was of analytical grade. The palladium target was of a purity >99.9999%, and was 16 mm × 9 mm × 1 mm. The polyimide fiber paper had a thickness of 30 µm. All reagents were analytically pure and could be used directly in the experiment without further processing. All solutions were configured with ultra-pure deionized water with a resistivity of 18.2 MΩ, following the principle of reconfiguration for each experiment.

An electrochemical workstation (CHI660, Chenhua, Shanghai, China), laser with wavelength of 450 nm (xTool D1 Pro, Maker factory Technology, Shenzhen, China), Q-Switch pulsed fiber laser with wavelength of 1064 nm(MFP-50X, Maxphotonics, Shenzhen, China), scanning electron microscope (Sigma 300, Carl Zeiss AG, Oberkochen, Germany), Raman analyzer (JobinYvon XploRA, HORIBA, Longjumeau, France), resistance meter (4349B, Agilent Technology, California, USA), magnetic stirrer (MS-3D, SUNNE, Shanghai, China), electronic balance (BSA224S, Sartorius, Gottingen, Germany), heating plate (DB-XAB, LICHEN, Shanghai, China), and mortar and pestle (AGATE, LICHEN, Shanghai, China) were used.

### 2.2. Experiment

#### 2.2.1. Preparation of LIG

The material selected in this work is polyimide fiber paper and because of its own pores, there may be a different absorption of radiation in laser irradiation. Therefore, the paper needs a flame retardant to pre-treat it in order to prevent excessive ablation of the thinner paper-based part. Therefore, the pre-treatment of polyimide fiber paper in this thesis was uniformly taken to be as follows: moistened in 0.1 mM sodium tetraborate solution for 10 min, followed by overnight drying under ambient conditions. Rectangles with a side length of 2 cm were drawn by AutoCAD and then the pre-processed polyimide fiber paper was patterned using an xTool D1 Pro laser. After studying different combinations of laser power and scanning speed, the optimal parameters were obtained, with the laser power selected at 22% (2.2 W) and the scanning speed set at 24 mm·s^−1^ (P22S24). The LIG flake was gently removed with tweezers and transferred to a mortar and pestle, and the powdered graphene was obtained after sufficient grinding and dried for storage.

#### 2.2.2. Preparation of Palladium Nanoparticles

PdNPs nanoparticles were prepared by laser ablation of metallic palladium targets in anhydrous ethanol solution by liquid-phase pulsed laser ablation. Palladium solid targets were sequentially ultrasonically cleaned in ultrapure water, acetone, and ultrapure water, and then blown dry with nitrogen or air. The metal target was fixed at the bottom of a glass container containing 6 mL of anhydrous ethanol solution and 3 mm below the liquid level. Then, a pulsed fiber laser (wavelength 1064 nm) was used to irradiate the metal target for 10 min. The laser energy, moving speed, and irradiating distance were set to 25 W, 5 mm·s^−1^, and 16.5 cm, respectively. The target was removed at the end of the experiment and the colloidal solution in the glass container was collected with a pipette gun.

#### 2.2.3. Synthesis of LIG/PdNP Composites

The electrode modification solution was obtained by placing 50 mg of LIG powder and 20 µL of colloidal solution containing palladium nanoparticles in a mortar and grinding thoroughly for 30 min. A mass of 25 mg of carbon slurry and 60 µL of solvent were added and fully ground for 2 h to obtain the electrode modification solution. As a comparison, composites without the addition of LIG or PdNPs were prepared under the same conditions.

#### 2.2.4. Preparation of Modified Electrodes

Surface modification of the working electrode was carried out on screen-printed carbon electrodes (SPE). A 10 µL drop of the mixture was applied to the surface of the working electrode and the electrode was allowed to dry at 50 °C under heating conditions. After the electrode surface was hardened, it was sonicated in deionized water for 10 min to remove electrode surface impurities. The modified screen-printed electrode was denoted as PdNPs/LIG/SPE. At the same time, modified electrodes without LIG and without PdNPs were prepared under the same conditions and expressed as PdNPs/SPE and LIG/SPE, respectively. The preparation of the mixed solution and the modification process of the electrode are shown in Figure 1.

### 2.3. Electrochemical Measurements

The electrochemical experiments were all carried out using a CHI660 electrochemical workstation (Shanghai, China) under a room temperature environment. The modified electrode was used as the working electrode, an Ag/AgCl electrode as the auxiliary electrode, and a carbon electrode as the counter electrode to form a three-electrode system. The detection performance of the electrodes was evaluated by cyclic voltammetry and chronoamperometry by adding different concentrations of hydrogen peroxide in 0.1 M PBS solution as the detection target. To investigate the electrical properties of the modified electrode, we also performed CV and electrochemical impedance spectroscopy experiments in a PBS solution containing 5 mM potassium ferricyanide. The parameters for cyclic voltammetry were set to potentials ranging from −0.6 to 0.6 V and −1.0 to 0.8 V. The effect of different scan rates was explored and the electroactive area of the electrodes was calculated. A constant potential of −0.7 V was used for the chronocurrent method. The frequency range in the electrochemical impedance experiments was set from 0.1 to 1 × 10^6^, and the data were analyzed and fitted using ZView software version 2.8.

### 2.4. Morphological and Structural Characterization

The resistance of LIG prepared with different laser powers and scanning speeds was characterized using an Agilent 4349B Resistance Meter in order to investigate the optimal parameters for the preparation of LIG. A Zeiss Sigma 300 scanning electron microscope (SEM) was used to study the surface morphology and microstructure of the samples. Raman spectra were collected using a HORIBA JobinYvon XploRA Raman spectrometer based on a laser wavelength of 532 nm with an exposure time of 30 s at 25 mW laser power.

## 3. Results and Discussion

### 3.1. Characterization of LIG and PdNPs

In order to explore the optimal parameters for preparing LIG, we set different combinations of laser power and scanning speed, with power ranging from 14% to 28% and scanning speed ranging from 16 mm·s^−1^ to 40 mm·s^−1^. The image of LIG resistance versus laser power and scanning speed was plotted to determine the optimal parameters used to prepare LIG, as shown in Figure 2. In a certain range, when the laser power is the same, the resistance value increases with an increase in scanning speed. A possible reason is that the radiation energy absorbed by the paper base per unit area is too little when the scanning speed is too fast, and the low energy is not enough to break and recombine the bond inside the polyimide fiber paper structure, which directly leads to a low LIG conversion rate. When the scanning rate is the same, the resistance value decreases with an increase in the laser power, which is due to the fact that enough energy reaches the inner part of the paper fibers, which leads to an increase in the synthesis rate of LIG. The results verified that the introduction of flame retardants can effectively reduce the thermal degradation of paper fibers caused by laser heat [31,32]. It is worth noting that too high a laser power can make the paper completely ablative and the resistance cannot be measured. According to the resistance value of LIG and evaluation of the actual sample, the optimal combination of parameters was concluded to be a laser power of 22% and a scanning speed of 24 mm·s^−1^.

Raman spectroscopy is often used to characterize the layers and surface defects of graphene due to its high resolution of the spatial structure of matter [33]. Figure 3a shows the three characteristic peaks D, G, and 2D of the resulting product located at 1352 cm^−1^, 1756 cm^−1^, and 2700 cm^−1^, respectively. The D peak is generated by the disordered vibration of graphene, and its appearance indicates that there are numerous defects in LIG; the G peak is induced by the stretching vibration between carbon atoms, which can indicate the successful synthesis of graphene; the 2D peak is a double phonon resonance second-order Raman peak affected by the wavelength of the excitation light, and its intensity indicates that graphene is multilayered [34,35].

The morphology of LIG and palladium nanoparticles are observed by scanning electron microscopy and the results are shown in Figure 3. SEM images at different magnifications (Figure 3b,c) show a loose and porous 3D mesh structure of LIG, which is attributed to the release of the products as gases during laser irradiation [36]. Laser-induced graphene prepared by a short-wave UV laser has a higher N content and faster charge transfer rate, which is attributed to the smaller spot size and higher resolution to avoid excessive ablation and thus increase the conversion rate of LIG [37]. The formation of this porous structure facilitates the provision of more active sites, which helps to reduce electrical resistance and facilitates the transfer of electrons. The liquid-phase pulsed laser ablation technique forms plasma through the interaction of the laser with the surface of the target material, and the cavitation bubbles burst with the change of temperature and pressure making the surface atoms of the nanoparticles partially charged [38], which is the reason why the colloidal solution is stable without the addition of active agents [39,40]. The SEM image of palladium nanoparticles prepared in anhydrous ethanol solution by liquid-phase pulsed laser ablation technology is shown in Figure 3d, and we can conclude that the nanoparticles are more uniformly dispersed and have a wider size distribution in the range of 10 nm to 40 nm [41,42,43].

### 3.2. Electrochemical Characterization of Modified Electrodes

The electrical properties of the bare electrode (SPE), the modified electrode (PdNPs/LIG/SPE), and the electrodes used as controls (PdNPs/SPE, LIG/SPE) are characterized by the Fe(CN)_6_^3−/4−^ redox probe. The charge transfer process occurring at the electrode surface was investigated using electrochemical impedance spectroscopy, yielding Nyquist plots as shown in Figure 4a,b. The semicircular portion of the high-frequency region of the Nyquist diagram reflects the electron-transfer-controlled process at the electrode surface and the linear portion of the low-frequency region represents the diffusion-controlled process in solution [44]. The modified electrode PdNPs/LIG/SPE has a smaller radius of the fitted circle at high frequencies, indicating a lower charge transfer resistance and faster electron transfer rate at the electrode–solution interaction interface. The charge transfer resistances (*R_CT_*) of bare SPE, PdNPs/SPE, LIG/SPE, and PdNPs/LIG/SPE were derived as 19,873 Ω, 10,547 Ω, 210.9 Ω, and 72.86 Ω based on a Randles equivalent circuit, suggesting that the incorporation of LIG and PdNPs led to the decrease of impedance and increase of electrical conductivity at the electrode surface.

To further investigate the kinetics of the reaction process and the surface properties of the electrodes, cyclic voltammetry experiments were performed on bare SPE, PdNPs /SPE, PdNPs/ SPE, and PdNPs/LIG/ SPE, in a supporting electrolyte solution (0.1 M PBS, pH = 7.0) of 5 mM K_3_[Fe(CN)_6_] with the voltage range set to −0.6 V to 0.6 V and a scan rate of 50 mV·s^−1^. As shown in Figure 4c, the electrodes have different catalytic effects on the redox probes, and the redox peak currents of the modified electrodes PdNPs/LIG/SPE and LIG/ SPE are much higher than those of SPE and PdNPs/SPE, and the voltage differences at the redox peak positions are smaller, which may be attributed to the fact that LIG has a good electrical conductivity conducive to the rapid transfer of electrons and acceleration of the catalytic process. The redox peak current of the modified electrode PdNPs/LIG/SPE is two times higher than that of LIG/SPE because of the synergistic effect of PdNPs and LIG, which could provide more active sites for the reactants. The electroactive surface area (ESA) of the different modified electrodes is evaluated by the Randles–Sevcik equation:(1)$$Ip=2.69\;\times \;10^{5}\cdot A\cdot D^{\left(1/2\right)}\cdot n^{\left(3/2\right)}\cdot v^{\left(1/2\right)}\cdot C,$$where Ip is the peak current, A is the electroactive surface area, D is the diffusion coefficient, n stands for the number of electrons involved in the charge transfer process, v represents the scan rate in the CV experiments, and C is the concentration of the supporting electrolyte. The maximum surface area of the modified electrode PdNPs/LIG/SPE is 0.665 cm^2^, which is much higher than that of SPE (0.118 cm^2^), PdNPs/SPE (0.123 cm^2^), and LIG/SPE (0.284 cm^2^). The electroactive surface area differs from the physical area in that the ESA represents the area that actually plays a role in the catalytic reaction process, which explains the maximum redox current of the modified electrode PdNPs/LIG/ SPE in potassium ferricyanide solution. Figure 5 shows the cyclic voltammogram images of the modified electrode PdNPs/LIG/SPE in potassium ferricyanide solution (0.1 M PBS, pH = 7.0) at different scan rates in the voltage range of −0.6 V to 0.6 V as a means to study the reaction-controlled processes at the electrode surface. The redox peak currents all increase with increasing scanning rate. This is because the faster the scanning rate, the greater the concentration gradient established over a shorter distance, leading to an increase in the diffusion flux, which is ultimately reflected in an increase in the current response as well. A linear relationship with a correlation coefficient of 0.9955 can be obtained by plotting the image of the reduction peak current versus the square root of the scan rate, as shown in Figure 5b. The result indicates that the electrochemical process occurring on the electrode surface is diffusion-controlled and the symmetrical redox peaks suggest that the process is quasi-reversible.

### 3.3. Electrochemical Response of Modified Electrodes to Hydrogen Peroxide

In order to investigate the electrocatalytic performance of the modified electrodes on hydrogen peroxide, the modified electrode PdNPs/LIG/SPE and the electrodes SPE, PdNPs/SPE, and PdNPs/SPE, which were used as the controls, were subjected to CV detection in a 1 mM H_2_O_2_ solution (0.1 M PBS, pH = 7.0) in the potential range of −1.0 V to 0.8 V at a scan rate of 50 mV·s^−1^. As shown in Figure 6, the current response to hydrogen peroxide varied greatly among different electrodes. The CV curves of the bare electrode SPE and the modified electrode PdNPs/SPE are highly overlapped, with no obvious redox current peaks observed, which could be attributed to the fact that the effective catalytic area is reduced by the agglomeration of PdNPs on the electrode surface, and that there is no conductive material in the modified substance that could provide and transfer the charge for the catalytic process in a timely manner. When the electrode surface is only modified with LIG, it is clearly seen that the background current response in the catalytic process increases significantly, but there is no obvious current peak, which is attributed to the fact that the excellent conductivity of LIG promotes electron transfer, but the catalytic performance for hydrogen peroxide is poor. According to the CV curve of the modified electrode PdNPs/LIG/SPE, there is an obvious reduction peak current near −0.7 V, which indicates that the modified electrode can promote the catalytic process of H_2_O_2_.

In order to analyze the catalytic ability of the modified electrode PdNPs/LIG/SPE for different concentrations of hydrogen peroxide, CV experiments were carried out by adding different concentrations of hydrogen peroxide to 0.1 M phosphate buffer (pH = 7.0), setting the voltage range from −1 V to 0.8 V, and using a scanning rate of 50 mV·s^−1^. As shown in Figure 7a, the modified electrode PdNPs/LIG/SPE has a good CV response in different concentrations of hydrogen peroxide solution. With an increase in hydrogen peroxide concentration, the oxidation peak is observed in the range of 0.189 V to 0.307 V with a slight increase in the peak current (∆*I* = 0.089 mA), and the reduction peak is observed in the range of −0.689 V to −0.823 V with a significant increase in the peak current (∆*I* = 0.392 mA). In Figure 7b, the following linear regression equation is obtained by plotting the value of reduction peak current versus hydrogen peroxide concentration:I = −0.097C(H_2_O_2_) − 0.156 (R^2^ = 0.9959),(2)

The good detection ability of the modified electrode PdNPs/LIG/SPE for H_2_O_2_ is attributed to the fact that the loose and porous structure of LIG provides more adsorption sites for PdNPs to avoid concentration saturation, and the good electrical conductivity of LIG can provide and transfer charges in time for the catalytic process of PdNPs, which greatly improves catalytic performance.

In addition, the electrochemical reaction of the modified electrode PdNPs/LIG/SPE with different concentrations of H_2_O_2_ was carried out by a chronocurrent (IT) experiment at a voltage setting of −0.7 V as a method to study the detection performance of the sensor at low concentrations. In performing IT experiments, we first added PBS solution to react for 100 s to reach a stable current, and then we added hydrogen peroxide solution every 20 s to achieve different concentration detection environments. As shown in Figure 7c, the current changes sharply at the moment of adding hydrogen peroxide, and then gradually slows down and reaches a steady state in the time period of 10 s, indicating that the sensor has a good response speed and a high saturation degree. Based on the current response versus hydrogen peroxide concentration, a calibration curve is obtained as shown in Figure 7d. The limit of detection (*LOD*) is calculated according to the following equation:*LOD* = 3*δ⁄k*, (*S⁄N* = 3),(3)
where *δ* represents the standard deviation of the blank responses and *k* is the slope of the calibration curve. The limit of detection of the modified electrode is calculated to be 0.37 µM with a sensitivity of 195.89 µA·µM^−1^·cm^−2^. Good linearity was observed in the concentration range of 5 µM to 0.9 mM and 0.9 mM to 5 mM. Considering the parameters, such as linear range, minimum detection limit, and sensitivity, the performance of the sensor prepared in this paper is better (Table 1).

### 3.4. Effect of Scan Rate

The CV experiments were carried out in PBS solution (0.1 M, pH = 7.0) containing 1 mM H_2_O_2_ with the voltage range set from −1 V to 0.8 V at different scan rates in order to investigate the effect of different scan rates on the performance of the modified electrode PdNPs/LIG/SPE in catalyzing hydrogen peroxide as shown in Figure 7e. According to the CV curves, it can be seen that with an increase in scanning rate, the values of the redox peak currents increase and the corresponding voltage values move to negative potentials. By drawing the reduction peak current versus the square root of the scan rate in Figure 7f, we obtain a linear equation with a correlation coefficient of 0.9873, which further verifies that the electrochemical reaction of H_2_O_2_ catalyzed by the modified electrode PdNPs/LIG/SPE is controlled by a diffusion process. Based on the response current versus background current in these experiments, we chose 50 mV s^−1^ as the optimal scan rate and applied it to all CV experiments.

### 3.5. Reproducibility, Stability, and Anti-Interference

An important indicator for evaluating the performance of sensors is reproducibility, so we verified the performance of the sensors by performing CV experiments in a solution containing 1 mM hydrogen peroxide. In order to investigate the reproducibility of the modified electrode PdNPs/LIG/SPE, we prepared five sensors in the same way and tested them in the same experimental environment. As shown in Figure 8a, the CV response curves of the five sensors are basically unchanged, and the relative standard deviation of the reduced peak current values was 5.4%, indicating that the preparation method in this paper is reproducible and can meet the demand of large-quantity preparation.

In order to examine the stability of the sensors, we stored the prepared modified electrodes PdNPs/LIG/SPEs in a 4 °C environment, and the CV measurements were performed every five days in a twenty-day range. As can be seen from Figure 8b, with the increase of days in storage, the peak current slightly decreases to the original at 96.06%, 92.74%, and 92.70%, which is acceptable. In addition, the cleaning behavior we performed on the sensors during the experiment did not have an effect on stability, which indicates that the sensors prepared in this paper are capable of performing multiple repetitive measurements.

A 100-fold concentration of AA, DA, UA, Glu, NaCl, or KCl was added to a hydrogen peroxide solution with a concentration of 5 µM and cyclic voltammetry was performed to verify the anti-interference of the sensor. As shown in Figure 8c, among the interfering substances, DA has the greatest effect on the peak current of hydrogen peroxide, which shows an increase of 0.73 µA. In general, the effect of the addition of common interfering substances on the peak current is less than 5.6%, indicating that the sensor prepared in this paper is highly specific to hydrogen peroxide.

## 4. Conclusions

In this work, we prepared PdNP/LIG composites for enzyme-free electrochemical detection of hydrogen peroxide by a laser technique. A series of material characterization and electrochemical experiments demonstrated that the modified electrodes PdNPs/LIG/SPEs have excellent catalytic activity for hydrogen peroxide. This is attributed to the fact that the loose and porous structure of LIG provides more binding sites for palladium nanoparticles, and the synergistic effect of them results in a substantial improvement in sensor performance. The PdNPs/LIG/SPE non-enzymatic hydrogen peroxide sensors prepared in this paper have a good linear response in the range of 5 µM to 0.9 mM and 0.9 mM to 5 mM. The limit of detection is 0.37 µM. From the stability test of the sensor, it can be seen that the sensor still has good responsiveness to hydrogen peroxide after repeated use and cleaning, which indicates that the non-disposable electrochemical sensor prepared in this paper can meet the demand of repeated use and reduce detection cost. Good reproducibility and stability provide the possibility of mass production, and, in addition, the excellent anti-interference ability means the sensor prepared in this paper can be applied in complex environments, which provides a new way of thinking for the preparation of electrochemical sensors in the future.

## Figures and Tables

**Figure 1 sensors-24-02043-f001:**
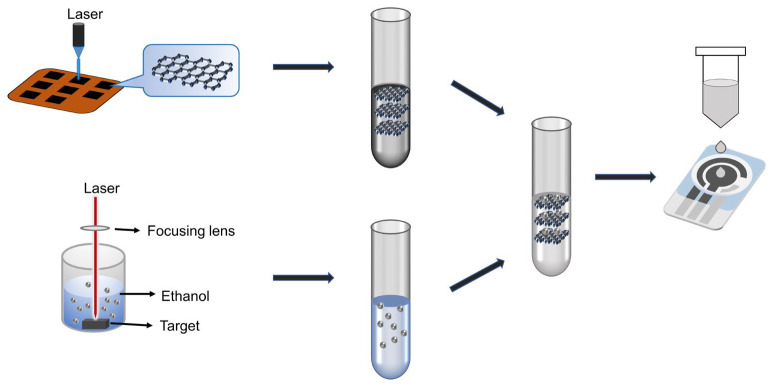
Schematic representation of material preparation and sensor modification.

**Figure 2 sensors-24-02043-f002:**
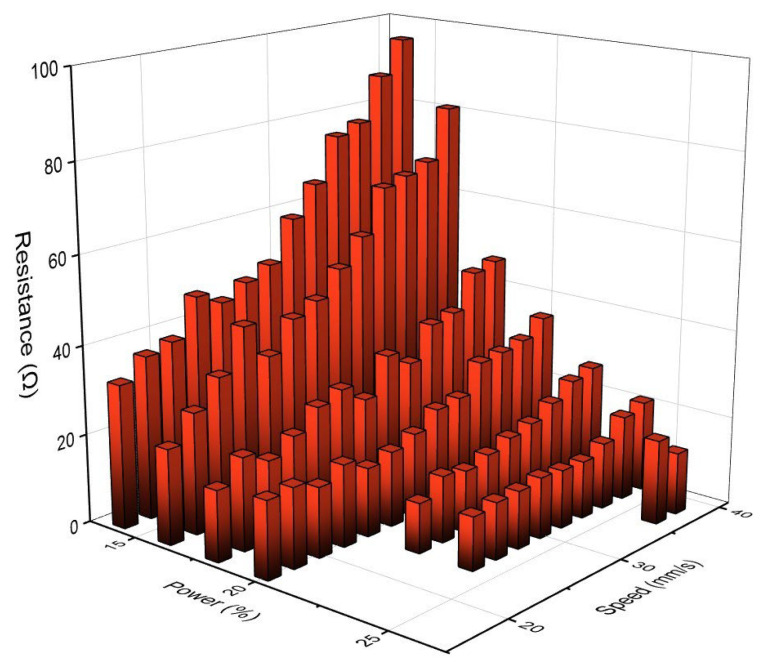
Resistance values of LIG prepared by different combinations of laser power and scanning speed.

**Figure 3 sensors-24-02043-f003:**
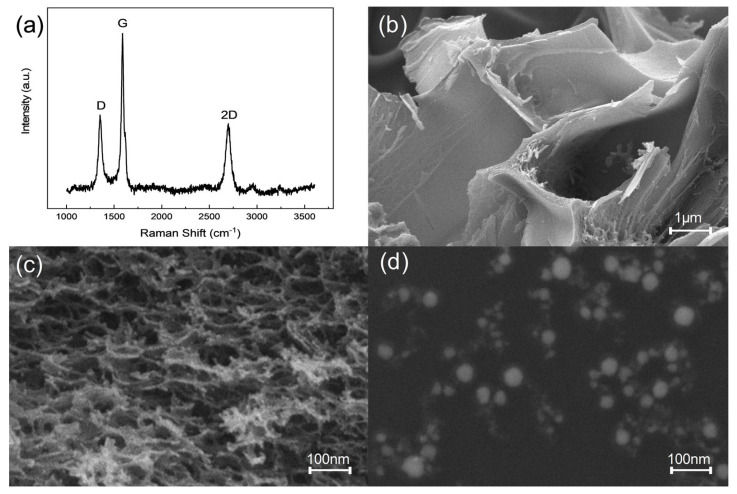
(**a**) Raman spectra of LIG produced by P22S24. SEM images of LIG with (**b**) lower magn-fication and (**c**) higher magnification. (**d**) SEM images of palladium nanoparticles prepared by pulsed laser ablation.

**Figure 4 sensors-24-02043-f004:**
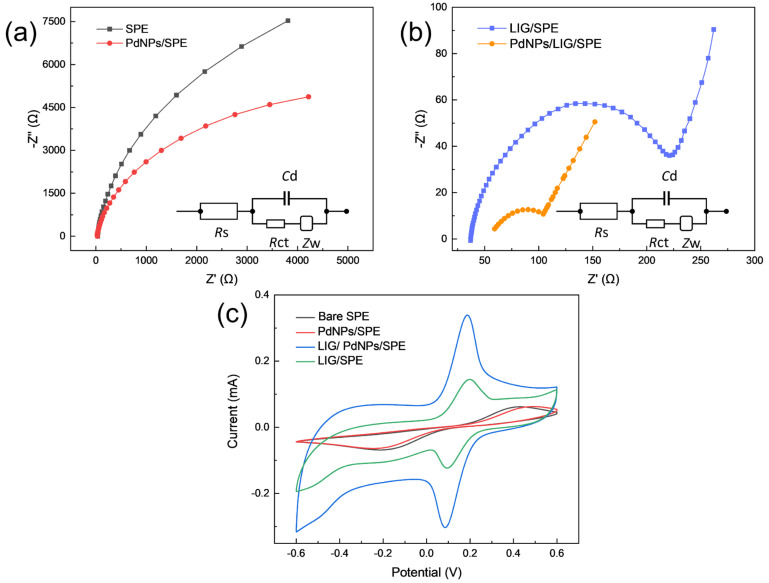
Nyquist plots obtained from impedance spectroscopy analysis for (**a**) SPE and PdNPs/SPE, (**b**) LIG/SPE and PdNPs/LIG/ SPE under study, with fitting of Nyquist plots with a Randles modified circuit (inset). Cyclic voltammograms obtained for (**c**) bare SPE, PdNPs/SPE, LIG/SPE, and PdNPs/LIG/ SPE with a scan rate of 50 mV/s, using 5 mM Fe(CN)_6_^3−^/Fe(CN)_6_^4−^ as a redox probe.

**Figure 5 sensors-24-02043-f005:**
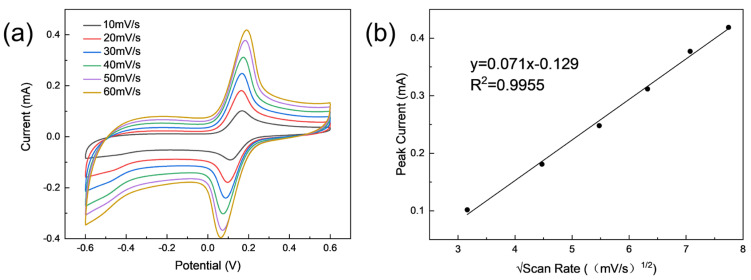
(**a**) Cyclic voltammograms obtained for the PdNPs/LIG/SPE with scan rates from 10 mV/s to 60 mV/s, using 5 mM Fe(CN)_6_^3−^/Fe(CN)_6_^4−^ as a redox probe; (**b**) the peak current as a function of the square root of the scan rate.

**Figure 6 sensors-24-02043-f006:**
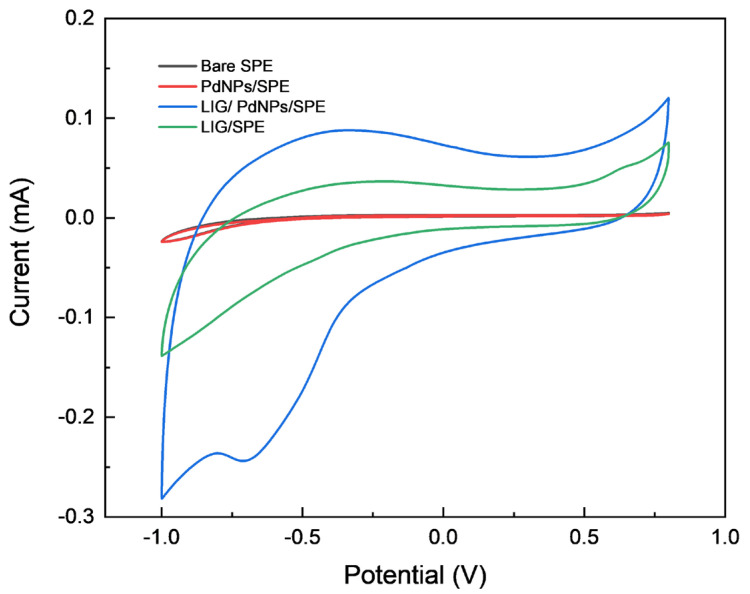
Cyclic voltammograms obtained for bare SPE, PdNPs/SPE, LIG/SPE, and PdNPs/LIG/SPE in the presence of 1 mM H_2_O_2_.

**Figure 7 sensors-24-02043-f007:**
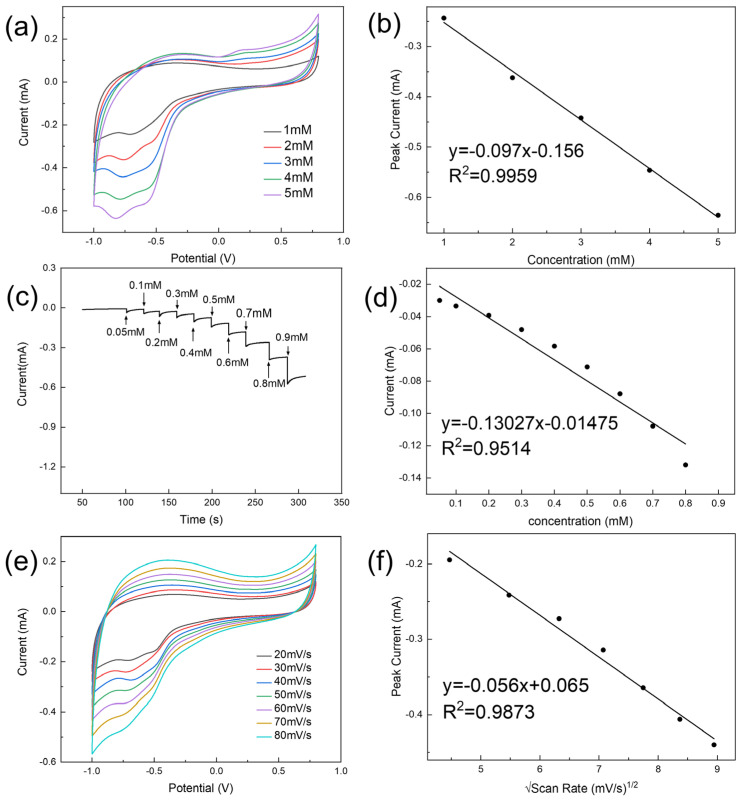
(**a**) CV curves with a scanning rate of 50 mV/s and (**c**) the chronocurrent response with an applied potential of -0.7 V of the PdNPs/LIG/SPE at different concentrations of H_2_O_2_; calibration curves of (**b**) high- and (**d**) low-concentration hydrogen peroxide with response currents. (**e**) CV curves of the PdNPs/LIG/SPE with different scan rates in the presence of 1 mM H_2_O_2_ and (**f**) the linear curve of the square root of the peak current and the scanning rate.

**Figure 8 sensors-24-02043-f008:**
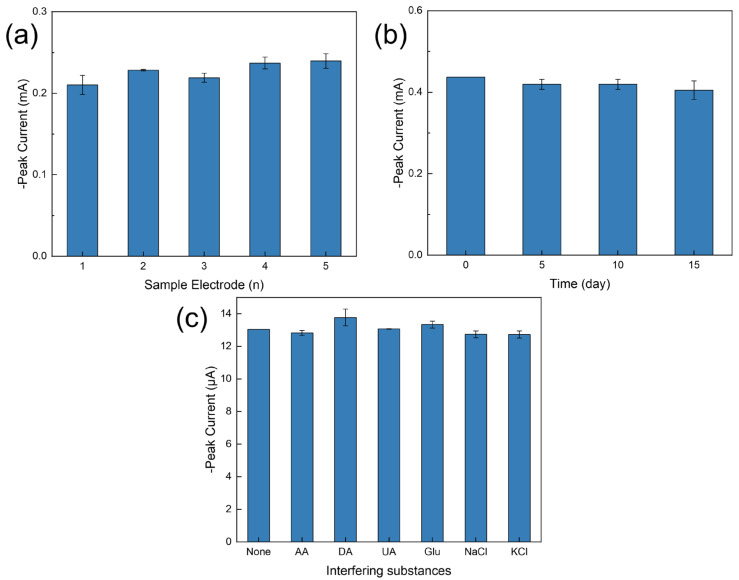
(**a**) The repeatability of five PdNPs/LIG/SPE sensors prepared under the same conditions by comparing the response peak currents. (**b**) The stability of the PdNPs/LIG/SPE sensor at ambient conditions for 15 days in 1 mM H_2_O_2_ with five-day intervals. (**c**) The anti-interference of PdNPs/LIG/SPE sensors with the addition of interfering substances to the H_2_O_2_ solution.

**Table 1 sensors-24-02043-t001:** Comparison of the analytical performance of the PdNPs/LIG/SPE with the previously reported literature for the detection of H_2_O_2_.

Modified Electrode	Linear Range	Detection Limit (μM)	Sensitivity (µA·µM^−1^·cm^−2^)	Ref
AuNPs/rGO ^1^/ITO ^2^	100 μM–500 μM	6.55	0.0641	[45]
PtNPs/LIG	50 μM–13.2 mM	11.6	16	[36]
Ag-CuNPs/GCE ^3^	2.0 mM–9.61 mM	152	----	[46]
Ni–Fe_3_O_4_/rGO/GCE	1 μM–1 mM	0.2	601.2	[47]
CuNPs/ERGO ^4^/Au	10 μM–10 mM	1.87	----	[48]
SnO_2_/AuNPs/GCE	49.98 μM–3937.21 μM	6.67	14.157	[49]
Ag-PtNPs/GE	0.2 μM–200 μM	0.12	----	[50]
Au-PtNPs/rGO	5 μM–400 μM	0.008	1117	[51]
N-rGO/ITO	100 μM–10.7 mM	26	305	[52]
PdNPs/LIG/SPE	50 µM–0.9 mM, 0.9 mM–5 mM	0.37	195.89	This work

^1^ rGO: reduced graphene oxide; ^2^ ITO: indium tin oxide electrode; ^3^ GCE: glassy carbon electrode; ^4^ ERGO: electrochemical reduction of graphene oxide.

## Data Availability

No new data were created.

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
