# Peer review of "A Non-Disposable Electrochemical Sensor Based on Laser-Synthesized Pd/LIG Nanocomposite-Modified Screen-Printed Electrodes for the Detection of H2O2"

_sensors, 2024, doi:10.3390/s24072043_

Round 1

Reviewer 1 Report

Comments and Suggestions for Authors

In this study, the authors discuss

" Non-disposable Electrochemical Sensor Based on Laser-Synthesized Pd/LIG Nanocomposites Modified Screen-printed Electrodes for the Detection of H2O2 "

The paper showed good results in relative major and I think needs some correction before final acceptance. I suggest that authors that more attention to questions and comments

1-Abbreviations in the abstract and introduction should be elaborated on their first appearance in the manuscript along with the abbreviation in the parentheses.

2-In the title, the author has used “non-disposable” for modified electrodes, so the author should give more details about this term.

3-In section “Reagents, Chemicals’’. All reagents need to be checked. It's important to mention the chemical purity.

4-In section 3.3: How the author did calculate LOD? Please give more details.

The lowest concentration of the linear range can not be selected as LOD. Please check again.

5-The author should delete the “lowest” or “minimum” for LOD in the manuscript.

6-In line 357, the author should correct the sentence as follows:

the reduction peak current versus the square root of the scan rate.

7. Some data should be confirmed with the error bar

Author Response

Thank you very much for your suggestions and the answers are attached.

Reviewer 2 Report

Comments and Suggestions for Authors

This manuscript is devoted to the development and research of the characteristics of a new electrochemical sensor for the hydrogen peroxide detection. This study is fully consistent with the subject of the journal. The experimental plan and the text of the article are quite well structured; the authors adhered to the standard plan for such a type of research devoted to the synthesis of the electrochemical sensors. After some improvements, I can recommend this work for publication.

 1) The novelty of this work must be eloborated in the end of the Introduction. Are the authors the first to propose using the laser direct writing technology and liquid phase pulsed laser ablation techniques for the synthesis of graphene composites with metal nanoparticles? How does the modified electrode proposed in this work differ from other recent studies? It is also necessary to explain why the authors chose Pd nanoparticles.

 I also advise you to generally reconsider the logic of presentation in the Introduction. Before the second paragraph, you need to add information explaining why the authors suddenly started talking about graphene. Now, the logic of the narrative is lost.

 2) There are inaccuracies when specifying optimal parameters for preparing LIG. Section 2.2.1 states that the laser power of 14% and the scanning speed of 24 mm/s (line 126) were selected. At the same time, in section 3.1, which describes the selection of these parameters based on the results presented in Fig. 2, the optimal parameters are indicated as follows: the laser power of 22% and the scanning speed of 24 mm/s (lines 200-202). Please, check this out.

 3) The authors conclude that prepared palladium nanoparticles “are more uniformly dispersed and have a wider size distribution”. What is this comparison being made with? Supplement these conclusions with references from the literature or experimental results.

 4) In my opinion, a comparative analysis of the data presented in Table 2 is required. The authors only indicated that the sensing characteristics of the prepared in this paper sensor are “excellent” than those previously reported in literature, while several electrodes according to the table have lower values of the limit of detection.

5) The article uses a large number of abbreviations, but there are some inaccuracies when using them:

- for several abbreviations their definitions are given twice: LIG (line 70 and line 75), SEM (line 83 and line 177), CV (line 91 and line 163), IT (line 91 and line 163), EIS (line 92 and line 166).

- several abbreviations are no longer used in the text (DI, PLAL, LOD) or are used only once (ESA, PI), they are not necessary.

- the abbreviation PdNPs is not deciphered. Also, under Table 2, you should decode all the abbreviations that are used in the names of the modified electrodes.

- the abbreviation LIG is used in the Abstract, but its definition is given only in the Introduction (line 70).

There are also some inaccuracies and typos in the article, for example:

1) lines 325: check, it is wrong.

2) Figs. 4c, 5a, 7a,c,e: it is necessary to increase the font size in the legend.

3) lines 379, Fig. 8c: the experiment with Glu is described, while in glutamic acid is not mentioned in section 2.1. Reagents, Chemicals, and Instruments.

4) Reconsider the structure of the article in relation to the location of the figures: move the figures to those sections where they are discussed (for example, Figs. 2, 4).

Author Response

Thank you for your advice. We apologize for our carelessness. Based on your comments, we have made changes and the answers are attached.

Round 2

Reviewer 1 Report

Comments and Suggestions for Authors

It can be accepted